# Causalized Convergent Cross Mapping and Its Implementation in Causality Analysis

**DOI:** 10.3390/e26070539

**Published:** 2024-06-24

**Authors:** Boxin Sun, Jinxian Deng, Norman Scheel, David C. Zhu, Jian Ren, Rong Zhang, Tongtong Li

**Affiliations:** 1Department of Electrical and Computer Engineering, Michigan State University, East Lansing, MI 48824, USA; sunboxin@msu.edu (B.S.); dengjinx@msu.edu (J.D.); renjian@msu.edu (J.R.); 2Department of Radiology, Michigan State University, East Lansing, MI 48824, USA; scheelno@msu.edu; 3Department of Radiology, Albert Einstein College of Medicine, Bronx, NY 10461, USA; zhuda@msu.edu; 4Institute for Exercise and Environmental Medicine, Texas Health Presbyterian Hospital Dallas, Dallas, TX 75231, USA; rongzhang@texashealth.org; 5Department of Neurology and Internal Medicine, University of Texas Southwestern Medical Center, Dallas, TX 75390, USA; 6Michigan Alzheimer’s Disease Research Center, Ann Arbor, MI 48109, USA

**Keywords:** causality, causalized convergent cross mapping, directed information

## Abstract

Rooted in dynamic systems theory, convergent cross mapping (CCM) has attracted increased attention recently due to its capability in detecting linear and nonlinear causal coupling in both random and deterministic settings. One limitation with CCM is that it uses both past and future values to predict the current value, which is inconsistent with the widely accepted definition of causality, where it is assumed that the future values of one process cannot influence the past of another. To overcome this obstacle, in our previous research, we introduced the concept of causalized convergent cross mapping (cCCM), where future values are no longer used to predict the current value. In this paper, we focus on the implementation of cCCM in causality analysis. More specifically, we demonstrate the effectiveness of cCCM in identifying both linear and nonlinear causal coupling in various settings through a large number of examples, including Gaussian random variables with additive noise, sinusoidal waveforms, autoregressive models, stochastic processes with a dominant spectral component embedded in noise, deterministic chaotic maps, and systems with memory, as well as experimental fMRI data. In particular, we analyze the impact of shadow manifold construction on the performance of cCCM and provide detailed guidelines on how to configure the key parameters of cCCM in different applications. Overall, our analysis indicates that cCCM is a promising and easy-to-implement tool for causality analysis in a wide spectrum of applications.

## 1. Introduction

Causality analysis aims to find the relationship between causes and effects by exploring the directional influence of one variable on the other, and it has been a central topic in science, economy, climate, and many other fields [1,2,3,4,5,6,7,8,9]. Compared with correlation, which reflects the mutual dependence between two variables, causality analysis may provide additional information since two time series with low correlation may have strong unidirectional or bi-directional causal coupling between them. Some representative examples can be found in [9].

The first practical causal analysis framework is Granger Causality (GC), which was proposed by Granger in 1969 [10]. GC is a statistical approach that relies on a multi-step linear prediction model and aims to determine whether the values of one time series are useful in predicting the future values of the other. As a well-known technique, the validity and computational simplicity of GC have been widely recognized [11,12,13,14]. At the same time, it has also been noticed that when there exists instantaneous and/or strong nonlinear interactions between two regions, GC analysis may lead to invalid results [9,15]. Moreover, GC may not be able to detect the causation in deterministic settings [10,16].

In 1990, directed information (DI)—the first causality detection tool based on information theory—was proposed by Massey [17] when studying discrete memoryless communication channels with feedback. DI measures the directed information flowing from one sequence to the other. As an information-theoretic framework, a major advantage of DI is that it is a universal method that does not rely on any model assumptions of the signals and is not limited by linearity or separability [18,19]. In refs. [9,18], the performance of DI in causality analysis was demonstrated using both simulated data and experimental fMRI data. It was found that DI is capable of detecting both linear and non-linear causal relationships. However, it was also noticed that the direct evaluation of DI relies heavily on probability estimation and tends to be sensitive to data length as well as the step size used in the quantization process [9].

In 2012, convergent cross mapping (CCM), a new causality model based on state space reconstruction was proposed by Sugihara et al. [16], and it was demonstrated that CCM could serve as an effective tool in addressing non-separable systems and identifying weakly coupled variables under deterministic settings, which may not be covered by GC. Since then, CCM has attracted considerable attention from the research community in many different fields [20,21,22,23,24,25,26,27,28].

Recall that causality aims to determine whether the current and past values of one time series are useful in predicting the future values of another in addition to its own past values. In CCM, however, both the past and future values are utilized to reconstruct the current value [9]. As a result, the causality defined by CCM is inconsistent with the original, widely accepted definition of causality where the key assumption is that the future values of one process cannot influence the past of the other.

Motivated by this observation, in [9], we introduced the concept of causalized convergent cross mapping (cCCM). More specifically, if only the current and historical values of *X* and the past values of *Y* are used to predict the current value Y(t), and vice versa, then CCM is converted to causalized CCM. We further proved the approximate equivalence of DI and cCCM under stationary ergodic Gaussian random processes [9].

This study is a continued work of our previous research [9] and is focused on the implementation perspective of cCCM in causality detection. More specifically, in this study, we aimed to further investigate the effectiveness of cCCM in identifying both linear and nonlinear causal coupling in various settings through a large number of examples, including Gaussian random variables with additive noise, sinusoidal waveforms, autoregressive models, stochastic processes with a dominant spectral component embedded in noise, deterministic chaotic maps, and systems with memory, as well as experimental functional Magnetic Resonance Imaging (fMRI) data. In particular, we analyze the impact of shadow manifold construction on the performance of cCCM and provide detailed guidelines on how to configure the key parameters of cCCM (especially the shadow manifold dimension and time lag) in different applications. Moreover, we examine the noise effect in cCCM and show that, in general, reliable causality detection can be achieved when the signal-to-noise ratio (SNR) is larger than 15 dB. Overall, our analysis indicates that CCM is a promising and easy-to-implement tool for causality analysis in a wide spectrum of applications.

The rest of the paper is organized as follows. In Section 2, we briefly revisit the original CCM, the causalized CCM (cCCM), the conditional equivalence between cCCM and DI, and the extension of bivariate cCCM to multivariate cCCM. In Section 3, we present the major results of the study, where we discuss the impact of noise on the performance of cCCM, evaluate the effectiveness of cCCM in causality analysis through numerous examples, provide detailed guidelines on the configuration of cCCM, and compare the performances of bivariate and multivariate cCCM and GC through both simulation examples and experimental fMRI data. Finally, we present the conclusions drawn from this research and provide related discussions in Section 4.

## 2. A Revisit of Causalized Convergent Cross Mapping

In this section, we first briefly revisit convergent cross mapping (CCM) [16] and introduce the concept of causalized CCM (cCCM) [9]. We then present the conditional equivalence of cCCM and the directed information framework [9] and introduce the extension of bivariate cCCM to multivariate cCCM.

**Convergent cross mapping (CCM).** The CCM algorithm is based on state-space reconstruction. Consider two dynamically coupled variables *X* and *Y* that share the same attractor manifold M. Let $X^{n}=\left[X_{1},X_{2},\cdots ,X_{n}\right]$ and $Y^{n}=\left[Y_{1},Y_{2},\cdots ,Y_{n}\right]$ be the time series consisting of samples of *X* and *Y*, respectively. The CCM causality analysis framework can be summarized as follows:**Step 1:** Construct the shadow manifolds with respect to $X^{n}$ and $Y^{n}$.
(1)$$\begin{array}{cc} M_{x} & =\{x_{t}\;|\;x_{t}=\left[X_{t},X_{t-\tau },\cdots ,X_{t-(E-1)\tau }\right],\;t=\left(E-1\right)\tau +1,\cdots ,n\}, \end{array}$$
(2)$$\begin{array}{cc} M_{y} & =\{y_{t}\;|\;y_{t}=\left[Y_{t},Y_{t-\tau },\cdots ,Y_{t-(E-1)\tau }\right],\;t=\left(E-1\right)\tau +1,\cdots ,n\}. \end{array}$$**Step 2:** For each vector $x_{t}$, find its E+1 nearest neighbors and denote the time indices (from closest to farthest) of the E+1 nearest neighbors of $x_{t}$ by $t_{1},\cdots ,t_{E+1}$.**Step 3:** If the two signals *X* and *Y* are dynamically coupled, then the nearest neighbors of $x_{t}$ in $M_{x}$ would be mapped to the nearby points of $Y_{t}$ on manifold M. The estimated $Y_{t}$ based on $M_{x}$, or say the cross mapping from *X* to *Y*, is defined as
(3)$${\hat{Y}}_{t}|M_{x}=\sum_{i=1}^{E+1}w_{i}Y_{t_{i}}$$
where
$$w_{i}=\frac{u_{i}}{\sum_{j=1}^{E+1}u_{j}},\;\;\mathrm{with}\;\;u_{i}=exp\left\{-\frac{d(x_{t},x_{t_{i}})}{d(x_{t},x_{t_{1}})}\right\},$$
where *d* denotes the Euclidean distance between two vectors. Please note that for every *i*, $\omega_{i}$ is a function of *t*. The cross mapping from *Y* to *X* can be defined in a similar way. As *n* increases, it is expected that ${\hat{X}}_{t}|M_{y}$ and ${\hat{Y}}_{t}|M_{x}$ would converge to $X_{t}$ and $Y_{t}$, respectively.**Step 4:** The cross mapping correlations are defined as
(4)$$\rho_{CCM}\left(X\to Y\right)=\rho \left(Y^{n},{\hat{Y}}^{n}\right)\;\;\;\mathrm{and}\;\;\;\rho_{CCM}\left(Y\to X\right)=\rho \left(X^{n},{\hat{X}}^{n}\right)$$
where ρ denotes the Pearson correlation.**Step 5:** If $\rho_{CCM}\left(X\to Y\right)>\rho_{CCM}\left(Y\to X\right)$ and converges faster than $\rho_{CCM}\left(Y\to X\right),$ then we say that the causal effect of *X* on *Y* is stronger than that in the reverse.

**Geometric illustration of convergent cross mapping.** Here, we provide the geometric illustration of convergent cross mapping from the shadow manifold $M_{x}$ to the shadow manifold $M_{y}$ under both strong and weak causal coupling.

Figure 1a corresponds to the situation when there is a strong causal relationship from *X* to *Y*, and Figure 1b illustrates the case when there is only a weak causation. For illustration purpose, the dimension of the shadow manifold was chosen to be E=2, the neighborhood of $x_{t}$ is represented using a the simplex consisting of three nearest neighbors, and the neighborhood of $y_{t}$ is represented in the same way.

**Causalized convergent cross mapping (cCCM).** Note that in CCM, both the past and future values are used in data reconstruction, which is inconsistent with the original definition of causality where it is assumed that the future values of one process cannot impact the past of another. For this reason, we propose the concept of causalized convergent cross mapping (cCCM).

More specifically, in CCM, if we limit the search of all the nearest neighbors in $M_{x}$ to $t_{i}<t$, i.e., we only use the current and previous values of *X* and the past values of *Y* to predict the current value $Y_{t}$, operating in the same way for the other direction, and then we obtained causalized CCM. That being said, **Step 2** in cCCM now becomes

**Step 2 for cCCM:** For each vector $x_{t}$, find its E+1 nearest neighbors in $M_{x}$ with an index smaller than *t* and denote the time indices (from closest to farthest) of the E+1 nearest neighbors of $x_{t}$ by $t_{1},\cdots t_{E+1}$. Note that for i=1,2,⋯,E+1, we now have $t_{i}<t$.

Then, we follow Steps 3–5 above, and denote the corresponding causalized cross mapping correlation, or the cCCM causation, as $\rho_{cCCM}$.

**Conditional equivalence between cCCM and directed information.** As an information-theoretic causality model, directed information (DI) measures the information flow from one time series to the other. DI plays a central role in causality analysis for two reasons. First, it is a universal method that does not have any modeling constraints on the sequences to be evaluated [29,30]. Second, DI serves as the pivot that links existing causality models GC [10,18], transfer entropy (TE) [9,31,32], and dynamic causal modeling (DCM) [33,34] through conditional equivalence between them.

In [9], we proved the conditional equivalence between cCCM and DI under Gaussian variables and used DI as a bridge to connect cCCM to other representative tools of causality analysis. More specifically, we showed that if (i) *X* and *Y* are dynamically coupled, zero-mean Gaussian random variables and their joint distribution is bivariate Gaussian, and (ii) $X^{n}$,$Y^{n}$ are stationary ergodic random processes; then, when *n* is sufficiently large,
(5)$$\begin{array}{c} {\bar{I}}_{n}\left(X\to Y\right)\approx -\frac{1}{2}\log\left(1-\rho_{cCCM}^{2}\left(X\to Y\right)\right), \end{array}$$
where ${\bar{I}}_{n}\left(X\to Y\right)$ denotes the average DI from *X* to *Y*, measured in bits per sample. The conditional equivalence of DI and cCCM under Gaussian random variables was demonstrated in [9] using experimental fMRI data.

This result also connects cCCM to other representative causality analysis frameworks in the family—GC, TE (Transfer Entropy, 2000 [31]), and DCM (Dynamic causal modelling, 2003 [33])—through the conditional equivalence between them under Gaussian random variables [9,12].

It is worth pointing out that the simulation-based analysis in [9] suggested that cCCM is often more robust in causality detection than DI. This is mainly because the DI calculation is based on probability estimation, which is sensitive to the step size used in the quantization process [35]. cCCM, on the other hand, gets around this obstacle through geometric cross mapping between the corresponding shadow manifolds, at the cost of a higher computational complexity. More specifically, cCCM relies on a K-nearest neighbor search and has a computational complexity of $O(n^{2})$ in the sequence length n, but the computational complexity of DI is only O(n).

**Extension of bivariate cCCM to multivariate cCCM.** Bivariate cCCM can be extended to multivariate conditional cCCM [9] based on a multivariate KNN search, which takes a similar approach as in the multivariate KNN predictability approaches [36,37,38,39].

Let $\Omega =\{X_{1},\cdots ,X_{L}\}$ denote the set of dynamically coupled random variables that share the same attractor manifold. As shown in [9], the multivariate conditional cCCM from $X_{j}\to X_{i}$ with respect to $\Omega \setminus \{X_{i},X_{j}\}$ (i.e., all the remaining random variables in Ω) is defined using the causality ratio as
$$\begin{array}{c} \mathrm{cCCM}\left(X_{j}\to X_{i}|\Omega \setminus \left\{X_{i},X_{j}\right\}\right)=\frac{\mathrm{Var}\left(e_{i}^{n}|\Omega \setminus \left\{X_{j}\right\}\right)-\mathrm{Var}\left(e_{i}^{n}|\Omega \right)}{\mathrm{Var}(e_{i}^{n}|\Omega \setminus \left\{X_{j}\right\})}, \end{array}$$
where $e_{i}^{n}|\Omega \setminus \left\{X_{j}\right\}$ denotes the estimation error vector based on $\Omega \setminus \{X_{j}\}$, and $e_{i}^{n}|\Omega$ is the estimation error vector based on the whole Ω. The definition can be adjusted by modifying Ω to reflect the conditional cCCM with respect to either an individual random variable or a group of random variables.

## 3. Results

### 3.1. The Impact of Estimation Error in cCCM

Note that CCM and cCCM are based on data reconstruction, and the reconstructed data converge to the true data as the data length goes to infinity when there exists causal coupling between the random variables under consideration. Here, we consider the impact of estimation error in cCCM.

As an example, we consider $\rho_{\mathrm{cCCM}}\left(X\to Y\right)=\rho \left(Y^{n},{\hat{Y}}^{n}\right)\approx \rho \left(Y,\hat{Y}\right)$. Note that
(6)$${\hat{Y}}_{t}|M_{x}=\sum_{i=1}^{E+1}w_{i}Y_{t_{i}}$$
where $t_{i}<t$, and
$$w_{i}=\frac{u_{i}}{\sum_{j=1}^{E+1}u_{j}},\;\;with\;\;u_{i}=exp\left\{-\frac{d(x_{t},x_{t_{i}})}{d(x_{t},x_{t_{1}})}\right\}.$$

When there exists estimation error, we can model $\hat{Y}$ as
(7)$$\hat{Y}=Y+n_{e}$$
where $n_{e}$ denotes the estimation error, which is independent of *Y*. In assuming $n_{e}$ is of zero-mean and variance $\sigma_{e}^{2}$, it can be shown that (please refer to the Supplementary file of [9])
(8)$$\rho \left(Y,\hat{Y}\right)=\frac{\sigma_{Y}}{\sqrt{\sigma_{Y}^{2}+\sigma_{e}^{2}}}.$$
where $\sigma_{Y}^{2}$ denotes the variance of *Y*. This result implies that the cCCM value $\rho_{\mathrm{cCCM}}\left(X\to Y\right)\approx \rho \left(Y,\hat{Y}\right)$ decreases as the estimation error power increases.

In the following, using the noise-free case as the benchmark, we examine the noise effect on cCCM through simulation examples, including Gaussian random variable and its signed and squared versions (**Examples 1** and **2**, respectively), as well as sinusoidal waveforms (**Example 3**). As shown in Table 1, when we increase the SNR from 0 dB to 20 dB, the cCCM value of the noisy signal gradually converges to the noise-free result. More specifically, our results suggest that reliable causality detection can be achieved when the SNR is larger than 15 dB.

The performance of cCCM is not only affected by noise but also closely related to the selection of *E* and τ. For **Examples 1** and **2** in Table 1, we chose E=5 and τ=1. For **Example 3**, we used E=5 and τ=5. Here, a larger τ is used mainly because X(t) and Y(t) are significantly over-sampled in **Example 3**. More discussion on the choice of shadow manifold parameters can be found in Section 3.2.

### 3.2. Causality Detection Using cCCM and the Choice of Shadow Manifold Parameters

In this section, we illustrate the performance of cCCM (together with CCM) in causality detection through simulation examples, including autoregressive models, stochastic processes with a dominant spectral component embedded in noise, deterministic chaotic maps, and systems with memory. As will be seen, these examples show that CCM and cCCM are sensitive to changes in coupling strength. It can be observed that CCM tends to result in larger causation values than cCCM; this is expected since CCM uses both the past and future values of *X* to predict the current value of *Y* (and vice versa), while cCCM only uses both the past values of *X* to predict the current value of *Y* (and vice versa).

We will also discuss the choice of key parameters—the dimension of shadow manifold *E* and the time lag τ—in the cCCM algorithm and the impact of these parameters on the detection of causal relationships. According to Takens’ theorem [40] and Whitney’s embedding theorem [41,42], the “magic number” is E=2d+1, and often less [16], where *d* is the dimension of the attractor M shared by *X* and *Y*. Another parameter, the time lag τ, is generally chosen as τ=1. When the signal is over-sampled, τ>1 can also be used.

It should be noted that for an accurate assessment of the causation, the sampling rate should always be chosen to be larger than the Nyquist rate. Otherwise, the causal relationship identified by cCCM may be invalid since the under-sampled sequences cannot capture the total information in the original signals.

#### 3.2.1. Examples on Autoregressive Models


**Example 4:**


Let *X* and *Y* be random processes given by
$$\begin{array}{cc} X(t+1) & =0.5X\left(t\right)+0.05Y\left(t\right)+n_{1}\left(t\right),\\ Y(t+1) & =0.65X\left(t\right)+0.08Y\left(t\right)+n_{2}\left(t\right), \end{array}$$
where $n_{1},n_{2}∼N\left(0,0.05^{2}\right),n_{1}$ and $n_{2}$ are independent, t=[0,1,2,⋯,2047], and X(0)=Y(0)=1.5. We chose E=5 and τ=1, and then the cCCM and CCM values between these two time series are
$$\begin{array}{cc} \; & \rho_{\mathrm{cCCM}}\left(X\to Y\right)=0.5067,\;\rho_{\mathrm{cCCM}}\left(Y\to X\right)=0.2210,\\ \; & \rho_{\mathrm{CCM}}\left(X\to Y\right)=0.5165,\;\rho_{\mathrm{CCM}}\left(Y\to X\right)=0.2294. \end{array}$$
The convergence of CCM and cCCM with respect to the data length is shown in Figure 2.


**Example 5:**


Let *X* and *Y* be random processes given by
$$\begin{array}{cc} X(t+1) & =0.6X\left(t\right)+0.3Y\left(t\right)+n_{1}\left(t\right),\\ Y(t+1) & =0.02X\left(t\right)+0.8Y\left(t\right)+n_{2}\left(t\right), \end{array}$$
where $n_{1},n_{2}∼N\left(0,0.05^{2}\right),n_{1}$ and $n_{2}$ are independent, t=[0,1,⋯,2047], and X(0)=Y(0)=1.5. Then, the cCCM and CCM values between these two time series are
$$\begin{array}{cc} \; & \rho_{\mathrm{cCCM}}\left(X\to Y\right)=0.3599,\;\rho_{\mathrm{cCCM}}\left(Y\to X\right)=0.5589\\ \; & \rho_{\mathrm{CCM}}\left(X\to Y\right)=0.4140,\;\rho_{\mathrm{CCM}}\left(Y\to X\right)=0.6222 \end{array}$$
The convergence of CCM and cCCM with respect to the data length is shown in Figure 2.

#### 3.2.2. Examples on Stochastic Processes with a Dominant Spectral Component


**Example 6:**


Let *X* and *Y* be two stochastic processes given by
$$\begin{array}{cc} X(t) & =0.1\sin\left(5\pi t\right)+0.6\sin\left(20\pi t\right)+n_{1}\left(t\right)\\ Y(t) & =0.6\sin\left(20\pi t\right)+n_{2}\left(t\right) \end{array}$$
where $n_{1},n_{2}$ are independent AWGN noise with SNR =10 dB, and t=0:0.005:2 (here, 0.005 is the step size). Then, the cCCM and CCM values between these two time series are
$$\begin{array}{cc} \; & \rho_{\mathrm{cCCM}}\left(X\to Y\right)=0.9314,\;\rho_{\mathrm{cCCM}}\left(Y\to X\right)=0.9175\\ \; & \rho_{\mathrm{CCM}}\left(X\to Y\right)=0.9362,\;\rho_{\mathrm{CCM}}\left(Y\to X\right)=0.9242 \end{array}$$
The convergence of CCM and cCCM with respect to the data length is shown in Figure 3.


**Example 7:**


Let *X* and *Y* be two stochastic processes given by
$$\begin{array}{cc} X(t) & =0.6\sin\left(5\pi t\right)+0.1\sin\left(20\pi t\right)+n_{1}\left(t\right)\\ Y(t) & =0.1\sin\left(20\pi t\right)+n_{2}\left(t\right) \end{array}$$
where $n_{1}$ and $n_{2}$ are independent AWGN noise with SNR =10 dB, and t=0:0.005:2. Then, the cCCM and CCM values between these two time series are
$$\begin{array}{cc} \; & \rho_{\mathrm{cCCM}}\left(X\to Y\right)=0.7108,\;\rho_{\mathrm{cCCM}}\left(Y\to X\right)=0.0616.\\ \; & \rho_{\mathrm{CCM}}\left(X\to Y\right)=0.7657,\;\rho_{\mathrm{CCM}}\left(Y\to X\right)=0.0517. \end{array}$$
The convergence of CCM and cCCM with respect to the data length is shown in Figure 3.

We selected τ=5 in **Examples 6** and **7** to reduce the impact of noise; please refer to Section 3.2.5 for more details.

#### 3.2.3. Examples on Deterministic Chaotic Maps


**Example 8:**


Let *X* and *Y* be two stochastic processes given by
$$\begin{array}{cc} X(t+1) & =X(t)[3.8-3.8X(t)],\\ Y(t+1) & =Y(t)[3.2-3.2Y(t)-0.1X(t)], \end{array}$$
where t=[0,1,⋯,2047], X(0)=0.7, and Y(0)=0.1. Then, the cCCM and CCM values between these two time series are
$$\begin{array}{cc} \; & \rho_{\mathrm{cCCM}}\left(X\to Y\right)=0.2164,\;\rho_{\mathrm{cCCM}}\left(Y\to X\right)=0.8923.\\ \; & \rho_{\mathrm{CCM}}\left(X\to Y\right)=0.1679,\;\rho_{\mathrm{CCM}}\left(Y\to X\right)=0.9705. \end{array}$$
The convergence of CCM and cCCM with respect to the data length is shown in Figure 4.


**Example 9:**


Let *X* and *Y* be two stochastic processes given by
$$\begin{array}{cc} X(t+1) & =X(t)[3.8-3.8X(t)-0.1Y(t)],\\ Y(t+1) & =Y(t)[3.2-3.2Y(t)-0.1X(t)], \end{array}$$
where t=[0,2047], X(0)=0.7, and Y(0)=0.1. Then, the cCCM and CCM values between these two time series are
$$\begin{array}{cc} \; & \rho_{\mathrm{cCCM}}\left(X\to Y\right)=0.8693,\;\rho_{\mathrm{cCCM}}\left(Y\to X\right)=0.9122.\\ \; & \rho_{\mathrm{CCM}}\left(X\to Y\right)=0.9704,\;\rho_{\mathrm{CCM}}\left(Y\to X\right)=0.9717. \end{array}$$
The convergence of CCM and cCCM with respect to the data length is shown in Figure 4.

#### 3.2.4. Examples on Systems with Memory

In this subsection, we examined the causal relationship in systems with memory (Examples 10–14) using CCM and cCCM under different choices of *E* and τ.


**Example 10:**


Consider a system with memory
$$\begin{array}{cc} \; & X=\mathrm{randn}(1024,1),\\ \; & Y(t)=0.2X(t-1)+0.85X(t-4). \end{array}$$
Here, (i) the MATLAB command “randn(1024,1)” returns an 1024-by-1 matrix of normally distributed random numbers; (ii) t=[0,1,⋯,1023] and X(t)=0 while t<0. The results corresponding to different *E* or τ values are displayed in Table 2.

We can see that the causation from X→Y cannot be fully captured when E=3 and τ=1.


**Example 11:**


Consider
$$\begin{array}{cc} \; & X=\mathrm{randn}(1024,1),\\ \; & Y(t)=0.85X(t-1)+0.85X(t-4). \end{array}$$
where t=[0,1,⋯,1023], and X(t)=0 while t<0. Then, for different *E* or τ values, the results are displayed in Table 3.

We can see that the causation from X→Y cannot be fully captured when E=3 and τ=1,2.


**Example 12:**


Consider a system with different dominant delays from **Example 11**:$$\begin{array}{cc} \; & X=\mathrm{randn}(1024,1),\\ \; & Y(t)=0.85X(t-2)+0.85X(t-4). \end{array}$$
where t=[0,1,⋯,1023], and X(t)=0 while t<0. Then, for different *E* or τ values, the results are displayed in Table 4.

We can see that the causation from X→Y cannot be fully captured when E=3 and τ=1.


**Example 13:**


Consider
$$\begin{array}{cc} \; & X=\mathrm{randn}(1024,1),\\ \; & Y(t)=0.8X(t-1)+0.8X(t-4)+0.6X(t-5) \end{array}$$
where t=[0,1,⋯,1023], and X(t)=0 while t<0. Then, for different *E* or τ values, the results are displayed in Table 5.

From this example, we can see the following: (i) when E=5,τ=1, we have x(t)=[X(t),X(t−1),⋯,X(t−4)], and the causation corresponding to item 0.6X(t−5) cannot be captured; (ii) when E=3 and τ=2, we have x(t)=[X(t),X(t−2),X(t−4)], and the causation corresponding to items 0.8X(t−1) and 0.6X(t−5) cannot be captured; and (iii) when E=6 and τ=1, we have x(t)=[X(t),X(t−1),⋯,X(t−5)], and the causation corresponding to all the items can be captured.

Now, if we consider the time-delayed causality, in which X(t) remains the same and $Y_{1}\left(t\right)=Y\left(t+1\right)$, then this is equivalent to considering the causality from $X_{1}\left(t\right)=X\left(t-1\right)$ to Y(t). In this case, as shown in Table 6, even when E=5 and τ=1, we have $x_{1}\left(t\right)=\left[X\left(t-1\right),X\left(t-2\right),\cdots ,X\left(t-5\right)\right],$ and the causation corresponding to all the items can be captured.


**Example 14:**


Consider
$$\begin{array}{cc} \; & X=\mathrm{randn}(1024,1),\\ \; & Y(t)=0.8X(t-4)+0.6X(t-5) \end{array}$$
where t=[0,1,⋯,1023], and X(t)=0 while t<0. Then, for different *E* or τ values, the results are displayed in Table 7.

In this example, both E=5,τ=1 and E=3,τ=2 can only capture the causation corresponding to 0.8X(t−4), and E=6 and τ=1 can capture the overall causation accurately.

Now, if we consider the time-delayed causality, in which X(t) remains the same and $Y_{3}\left(t\right)=Y\left(t+3\right)$, then this is equivalent to considering the causality from $X_{3}\left(t\right)=X\left(t-3\right)$ to Y(t). In this case, as shown in Table 8, E=5 and τ=1 work even better than E=6 and τ=1 since E=5 leads to a manifold with a lower dimension and, hence, a higher nearest neighbor density.

From Examples 10–14, it can be seen that in systems with memory, the selection of the shadow manifold dimension *E* and the signal lag τ largely rely on the positions of the dominant delays in the channel impulse response.

It can be seen that in systems with memory, for the accurate evaluation of CCM and cCCM causality, the following conditions need to be satisfied:(a)$E\cdot \tau >d_{d,max}$, where $d_{d,max}$ denotes the largest dominant delay.(b)For each *t*, the shadow manifold constructing vector x(t)=[X(t),X(t−τ),⋯,X(t−(E−1)τ)] should contain all the samples corresponding to the dominant delays.

It is also observed that if the conditions above are not satisfied, time-delayed cCCM from X(t−τ) to Y(t) might still capture the causation accurately if the instantaneous information exchange between X(t) and Y(t) is not significant. More specifically, if we consider a linear time-invariant (LTI) system $Y\left(t\right)=X\left(t\right)*h\left(t\right)=\sum_{l=0}^{L}h\left(l\right)X\left(t-l\right)$, where h(t) denotes the channel impulse response, when h(0) is negligibly small, we say that there is no significant instantaneous information exchange between X(t) and Y(t).

In the following two examples, we compare the performance between cCCM and Granger causality (GC) for systems with memory.


**Example 15:**


Consider a system with memory:(9)$$\begin{array}{cc} \; & X=\mathrm{randn}(1024,1), \end{array}$$(10)$$\begin{array}{cc} \; & Y(t)=0.8X(t)+0.2X(t-1)+0.2X(t-2)+0.2X(t-5)+n(t), \end{array}$$
where t=[0,1,⋯,1023], and X(t)=0 while t<0. Here, we assume that $n\left(t\right)∼N\left(0,\sigma^{2}\right)$ is independent of *X*. We then compare the performances of GC and CCM under different noise powers, and the results are shown in Table 9.

As can be seen, as long as the signal-to-noise ratio (SNR) is not too small, cCCM can capture the strong bidirectional causality between X(t) and Y(t), but GC cannot. This is mainly because cCCM takes the instantaneous information exchange between X(t) and Y(t) into consideration, but GC does not. That is, when there exists instantaneous information exchange between X(t) and Y(t), GC may fail to capture the causal coupling between X(t) and Y(t).

It is also observed that the $\rho_{\mathrm{cCCM}}$ value decreases as the noise power increases, which is consistent with our analysis in Section 3.1. When $\sigma^{2}=4$ and SNR =−7.74 dB, both cCCM and GC can no longer deliver valid results due to the strong noise effect.

Recall that the most commonly used method in Granger causality [10,11,12] analysis is to compare the following two prediction errors $e_{i}$ and $\tilde{e_{i}}$:$$Y_{i}=\sum_{j=1}^{K}a_{j}Y_{i-j}+e_{i}$$
$$\;\;\;\;Y_{i}=\sum_{j=1}^{K}b_{j}Y_{i-j}+\sum_{j=1}^{L}c_{j}X_{i-j}+{\tilde{e}}_{i},$$

And the Granger causality is defined to be the log-likelihood ratio
(11)$$\begin{array}{c} GC\left(X\to Y\right)=\ln\frac{\left|\mathrm{cov}\left(e\right)\right|}{|\mathrm{cov}\left(\tilde{e}\right)|}, \end{array}$$
where $e={\left[e_{1},e_{2},...,e_{n}\right]}^{T}$, $\tilde{e}={\left[{\tilde{e}}_{1},{\tilde{e}}_{2},...,{\tilde{e}}_{n}\right]}^{T}$, and |cov(·)| stands for the determinant of the covariance matrix.

Our results in Table 9 and the definition of GC suggest that the small fluctuations in the GC values as the noise variance increases from $10^{-6}$ to 4 are more likely to reflect the impact of the noise rather than the detection of the causality.


**Example 16:**


Consider
$$\begin{array}{cc} \; & X=\mathrm{randn}(1024,1),\\ \; & Y(t)=0.8X(t-1)+0.2X(t-2)+0.2X(t-5)+n(t), \end{array}$$
where t=[0,1,⋯,1023],X(t)=0 while t<0, and $n\left(t\right)∼N\left(0,10^{-6}\right)$ is an independently generated Gaussian noise. Then, the Granger causality between *X* and *Y* is
$$\begin{array}{c} GC(X\to Y)=11.3901,\\ GC(Y\to X)=0.0002. \end{array}$$
and the causality detected by cCCM is
$$\begin{array}{c} \rho_{\mathrm{cCCM}}\left(X\to Y\right)=0.9138,\\ \rho_{\mathrm{cCCM}}\left(Y\to X\right)=0.0128. \end{array}$$

In this example, there is no instantaneous information exchange between X(t) and Y(t), and both GC and cCCM detect the strong unidirectional causality from *X* to *Y* and deliver consistent results.

#### 3.2.5. Additional Examples on the Selection of the Dimension of the Shadow Manifold *E* and Time Lag τ

In this subsection, we illustrate the impact of *E* and τ on the performance of cCCM through some additional simulation examples, including single-tone time series embedded in noise (**Example 13**) and Gaussian stochastic process (**Example 14**). As in **Example 3**, it was found that a large E·τ value may help enhance the performance of cCCM under noise. However, it is also noticed that if *E* is too large, cCCM may no longer deliver valid results, as the excessively high dimension of the shadow manifold significantly reduces the density of the nearest neighbors, leading to inaccurate state-space reconstruction and causality evaluation.


**Example 17:**


This is a revisit of Example 3, with additional discussions on the selection of *E* and τ and different sampling instants. Consider the following noisy single-tone time series:$$\begin{array}{c} X\left(t\right)=\sin\left(t\right)+n_{1}\left(t\right),\\ Y\left(t\right)=\cos\left(t\right)+n_{2}\left(t\right), \end{array}$$
where t=0:0.01π:2π, and $n_{1}\left(t\right)$ and $n_{2}\left(t\right)$ are independent AWGN noises with SNRs varying in 0, 5, 10, 15, and 20 dB, or equal to 0 for all *t* in the noise-free case. By changing the values for *E* and τ, we are able to observe different noise effects. The simulation results for E=5,τ=1 and E=5,τ=5 are shown in Table 10 below.

As can be seen, as we increase the length of the data span E·τ, the noise effect is reduced. In particular, compared with E=5 and τ=1, when we choose E=5 and τ=5, a much better noise immunity is achieved, since E·τ is sufficiently long.

Note that increasing τ leads to the downsampling of the time series, and increasing *E* expands the dimension of the shadow manifolds. An over-increase in *E* or τ might downgrade the performance of cCCM. From Example 14, it can be seen that if *E* is much larger than 2d+1, cCCM may deliver inaccurate results.


**Example 18:**


Consider
$$\begin{array}{c} X=\mathrm{randn}(1024,1),\;\;Y=|X|, \end{array}$$

In this example, there is a strong unidirectional causality from *X* to *Y*, but very weak causation in the inverse direction. Choose τ=1. From Figure 5, we can see that as *E* increases, the cCCM value keeps on decreasing and reduces to 0.2 when E=50, which no longer reflects the strong unidirectional causality from *X* to *Y*.

#### 3.2.6. Examples of the Impact of Sampling Frequency on cCCM

In this subsection, we show that for the accurate assessment of causation, signals under consideration should be sampled with a sampling frequency higher than the Nyquist rate.


**Example 19:**


Consider
$$\begin{array}{cc} X & =\sin(k\pi t),\\ Y & =\cos(k\pi t), \end{array}$$
where t=0:0.005:4, and k=150,200,400. From Figure 6, it can be seen that if the sampling frequency is higher than the Nyquist rate, then strong bidirectional causal coupling can be detected between *X* and *Y*. On the other hand, if the sampling frequency is lower than the Nyquist rate, then the resulted cCCM value is no longer valid.

#### 3.2.7. Examples on Data Repetition in Causality Analysis

The following examples illustrate that even if *X* and *Y* are two independent signals that are not causally coupled, a causal pattern can be enforced in the concatenated time series through data repetition.


**Example 20:**


Let X= randn(1000, 1) and Y= randn(1000, 1) be two independent normally distributed time series. We have
$$\rho_{\mathrm{cCCM}}\left(X\to Y\right)=0.0198,\;\;\;\;\;\rho_{\mathrm{cCCM}}\left(Y\to X\right)=-0.0326.$$

That is, *X* and *Y* are not causally coupled. Consider
$$\tilde{X}=\left[X;X;X\right]\;\;\;\;\;\tilde{Y}=\left[Y;Y;Y\right],$$

Then, we have
$$\rho_{\mathrm{cCCM}}\left(\tilde{X}\to \tilde{Y}\right)=0.7860,\;\;\;\;\;\rho_{\mathrm{cCCM}}\left(\tilde{Y}\to \tilde{X}\right)=0.7810.$$
As can be seen, data concatenation results in strong causality that does not exist in the original *X* and *Y*.

#### 3.2.8. An Example of Multivariate Conditional cCCM


**Example 21:**


Let $X_{0}=$ randn(1024,1), $Y_{0}=$ randn(1024,1), and Z= randn(1024,1) be independent and normally distributed Gaussian random variables. Consider
$$\begin{array}{cc} X & =0.7X_{0}+10Z,\\ Y & =0.4Y_{0}+12Z. \end{array}$$
Then, the bivariate cCCM between *X* and *Y* is
$$\begin{array}{c} \rho_{\mathrm{cCCM}}\left(X\to Y\right)=0.9574,\;\;\;\rho_{\mathrm{cCCM}}\left(Y\to X\right)=0.9601, \end{array}$$
which provides a delusion that there exists strong bidirectional causality between *X* and *Y*. On the other hand, the multivariate cCCM between *X* and *Y* conditioning on *Z* is
$$\begin{array}{c} \mathrm{cCCM}(X\to Y|Z)=0.0298,\;\;\;\mathrm{cCCM}(Y\to X|Z)=0.0306, \end{array}$$
which accurately reflect the independent relationship between *X* and *Y*. From this example, it can be seen that conditional cCCM can help inspect the dependence among the random variables under consideration and may deliver more accurate results in the causality evaluation.

### 3.3. Application of cCCM for Brain Causality Analysis Using Experimental fMRI Data

In this study, we applied both bivariate and multivariate cCCM for a causality analysis of the brain network using experimental fMRI data and compared the results with those of GC [10,43].

We considered an fMRI dataset where fourteen right-handed healthy college students (7 males and 7 females, 23.4 ± 4.2 years of age) from Michigan State University volunteered to participate in a task-driven fMRI-based study. For each subject, fMRI datasets were collected on a visual stimulation condition with a scene–object fMRI paradigm, where each volume of images was acquired 192 times (8 min) while a subject was presented with 12 blocks of visual stimulation after an initial 10 s resting period. In a predefined randomized order, the scenery pictures were presented in six blocks, and the object pictures were presented in another six blocks. All pictures were unique. In each block, ten pictures were presented continuously for 25 s (2.5 s for each picture), followed with a 15 s baseline condition (a white screen with a black fixation cross at the center). The subject needed to press their right index finger once when the screen was switched from the baseline to the picture condition. More details on fMRI data acquisition and preprocessing can be found in [44].

Region of Interest (**ROI**) selection: we selected 10 ROI regions, including the left primary visual cortex (LV1), left parahippocampal place area (LPPA), left sensory motor cortex (LSMC), left parahippocampal white matter (LPWM), left retrosplenial cortex (LRSC), right primary visual cortex (RV1), right parahippocampal place area (RPPA), right sensory motor cortex (RSMC), right frontal white matter (RFWM), and right retrosplenial cortex (RRSC).

#### 3.3.1. Results for Bivariate and Multivariate cCCM

Note that the total length of the fMRI BOLD time series under visual stimulation condition was n=192, with the sampling period being 2.5 s. In the literature, it was reported that increasing the sampling rate of the fMRI signal can improve the robustness of the causality analysis [45]. Here, we first interpolate the fMRI sequence by a factor of 2 using the spline interpolation command in MATLAB and then conducted causality analysis for all the possible unidirectional regional pairs using both bivariate and multivariate cCCM.

The causality analysis results based on bivariate cCCM (averaged over all 14 subjects) are shown in Figure 7. Our results suggest the presence of unidirectional causality from LV1 → LSMC, RV1 → LSMC, LV1 → LPWM, LV1 → RFWN, and LPPA → LPWM.

The results corresponding to multivariate conditional cCCM with respect to individual brain regions are shown in Figure 8. As can be seen, RV1 has the most significant impact on the conditional causality from LV1 → LSMC, LV1→LPWM, and LV1→RFWM. This implies that RV1 has the highest inter-region dependence with LV1. For the same reason, LV1 has the most significant impact on the conditional causality from RV1 → LSMC. That is, multivariate conditional cCCM with respect to individual regions can detect unidirectional causality and also reflect the impact of interdependence between the ROIs on the conditional causality.

#### 3.3.2. Results for Bivariate and Multivariate Granger Causality (GC)

For comparison purposes, we analyzed the brain network causality using both bivariate and multivariate GC [12] with the same fMRI dataset.

From Figure 7 and Figure 9, it can be seen that bivariate GC delivers similar results as cCCM except for the causal coupling from LV1→LSMC. More specifically, cCCM shows that there exists unidirectional causality from LV1→LSMC, while GC shows that there exists bidirectional causality between LV1 and LSMC but no significant unidirectional causality. In ref. [9], the DI-based causality analysis also verified the presence of unidirectional causality from LV1→LSMC for the same dataset, which is consistent with the results of cCCM. These results suggest that for this fMRI dataset, cCCM tends to deliver a more accurate causality evaluation than GC.

A natural question arises: how should we explain the difference between cCCM and GC for the causality analysis here? Since based on the central limit theorem, fMRI signals can be modeled as Gaussian random variables for which cCCM and GC are conditionally equivalent. The underlying argument is that the equivalence between cCCM and GC under Gaussian random variables is subject to two conditions: (i) both X(t) and Y(t) follow the linear auto-regression model; and (ii) there is no significant instantaneous information exchange between X(t) and Y(t). More specifically, cCCM takes the instantaneous information exchange between X(t) and Y(t) into consideration, but GC does not. For this reason, when there exists instantaneous information exchange between X(t) and Y(t), GC may fail to capture the causal coupling between X(t) and Y(t), but cCCM succeeds. This is demonstrated through simulations in Example 15. In addition, cCCM can capture both linear and nonlinear causal causal coupling, but GC may have difficulty in detecting nonlinear causality. For these reasons, cCCM might be a more robust causality analysis tool than GC.

In the multivariate case, the theoretical relationship between cCCM and GC is not clear yet. In comparing Figure 8 and Figure 10, it can be seen that the results of multivariate cCCM and GC are largely consistent for LV1→LSMC and LV1→RFWM. However, they deliver very different results for the conditional causality from RV1→LSMC and LV1→LPWM with respect to other individual regions. In particular, for these two region pairs, the results of multivariate cCCM with respect to other individual regions are consistent with their bivariate counterparts and also reflect the impact of inter-region dependence on the conditional causality. However, the corresponding results of multivariate GC with respect to other individual regions vary significantly with the region under consideration, and 50% or more are no longer consistent with those of the bivariate GC.

Further theoretical analysis is needed on the theoretical relationship between conditional GC and multivariate cCCM, as well as the relationship between DI and the recent minimum entropy framework [46] in both bivariate and multivariate scenarios.

## 4. Conclusions and Discussion

In this paper, we revisited the definition of original CCM, identified the gap between CCM and the traditional definition of causality, presented causalized CCM (cCCM), and discussed the conditional equivalence of cCCM and directed information and the extension of bivariate cCCM to multivariate cCCM. We then evaluated the effectiveness of cCCM in the detection of causality through a large number of examples including Gaussian random variables with additive noise, sinusoidal waveforms, autoregressive models, stochastic processes with a dominant spectral component embedded in noise, deterministic chaotic maps, and systems with memory, as well as experimental fMRI data. We also examined the impact of noise on the performance of cCCM, and our results suggest that, in general, reliable results can be achieved when SNR >15 dB. In particular, we provided detailed discussions on the choice of the dimension of the shadow manifolds *E* and the time lag τ and the impact of these parameters on the detection of causal relationships using cCCM. Finally, we applied both bivariate and multivariate cCCM for the causality analysis of the brain network using experimental fMRI data and compared the results with those of GC.

Based on the conditional equivalence of cCCM and DI [9], we can see that cCCM provides an alternative way to evaluate the directed information transfer between stationary ergodic Gaussian random variables. Compared with DI, which relies heavily on probability estimation and tends to be sensitive to data length and quantization step size, cCCM, on the other hand, gets around this problem through geometric cross mapping between the manifolds involved.

However, the advantage of cross-mapping-based causality detection techniques comes with prices. The major limitation with CCM and cCCM is that they are based on the K-nearest neighbor (KNN) search algorithm and hence have a high computation complexity $O(n^{2})$, where *n* is the data length. The convergence speeds of CCM and cCCM also vary with the signals under applications and need to be taken into consideration in causality analysis, especially in dynamic systems where the causal relationships are time-variant. It is worthy to point out that when combined with the sliding window approach [47,48], cCCM can be used to evaluate time-varying causality in dynamic networks such as brain networks [49].

Overall, both our theoretical [9] and numerical analysis demonstrated that cCCM is a promising and easy-to-implement tool for causality detection in a wide spectrum of applications. In this paper, we showed that appropriate choices of *E*, τ, and the sampling frequency are critical for cCCM-based causality analysis and provided detailed guidelines on the configuration of cCCM. We wish that this paper can serve as a helpful reference on the implementation of cCCM for causality detection in different applications.

## Figures and Tables

**Figure 1 entropy-26-00539-f001:**
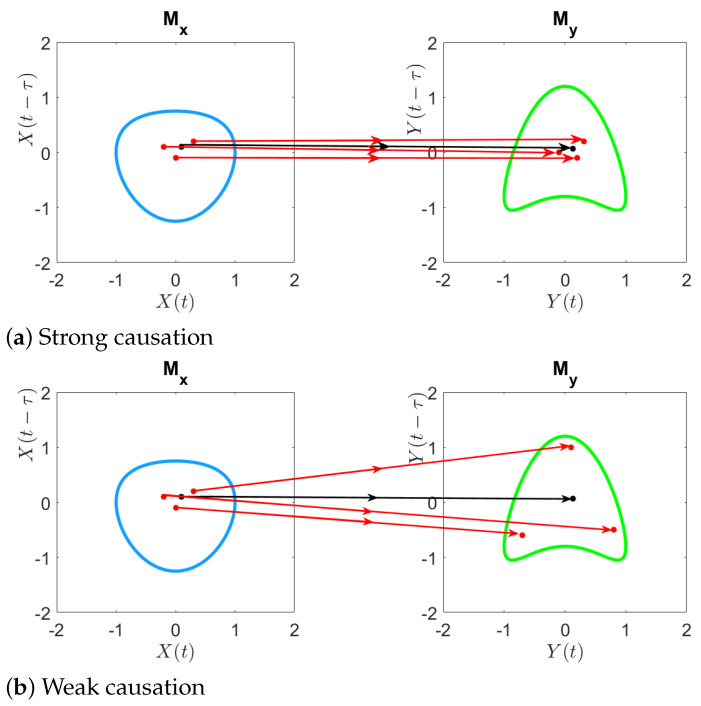
**Geometric illustration of the cross mapping from** $M_{x}$ **to** $M_{y}$. (**a**) When strong causation exists from *X* to *Y*, the nearest neighbors of $x_{t}$ are mapped to the nearest neighbors of $y_{t}$. (**b**) When there is only weak causation from *X* to *Y*, the nearest neighbors of $x_{t}$ are no longer mapped to the nearest neighbors of $y_{t}$.

**Figure 2 entropy-26-00539-f002:**
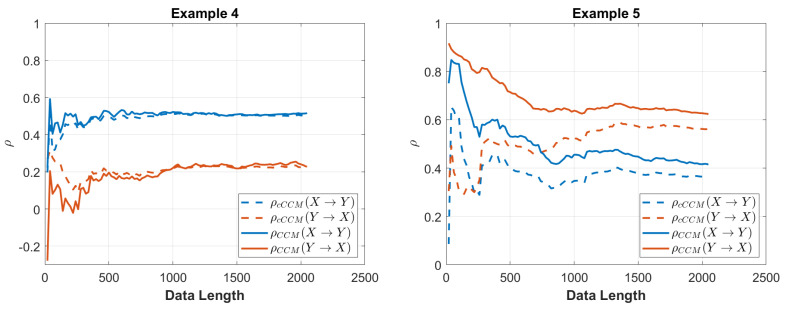
Performance of cCCM and CCM versus the data length for Examples 4 and 5.

**Figure 3 entropy-26-00539-f003:**
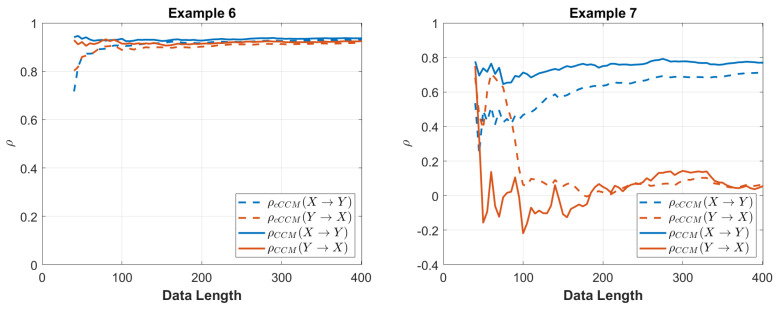
Performance of cCCM and CCM versus the data length for Examples 6 and 7.

**Figure 4 entropy-26-00539-f004:**
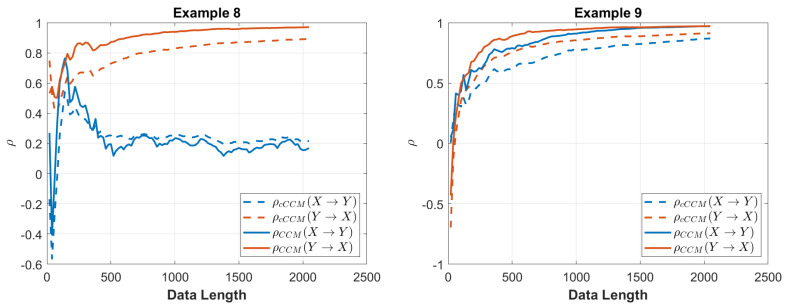
Performance of cCCM and CCM versus the data length for Examples 8 and 9.

**Figure 5 entropy-26-00539-f005:**
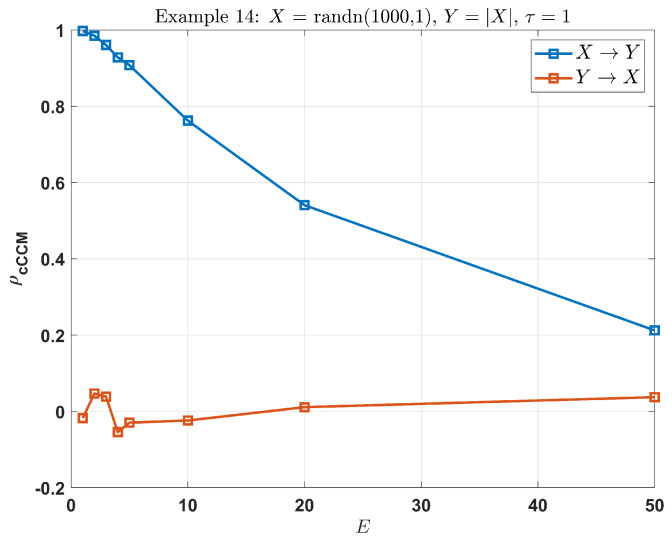
cCCM results for **Example 18** (X=randn(1024,1), Y=|X|): an excessively large *E* may downgrade the performance of cCCM; here, τ=1.

**Figure 6 entropy-26-00539-f006:**
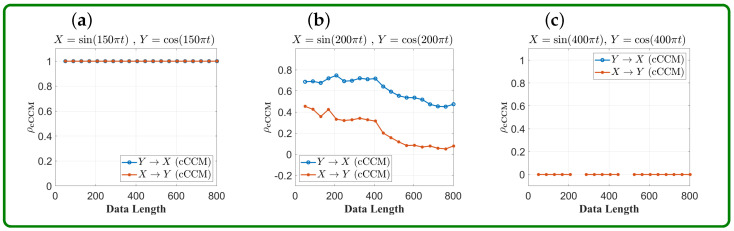
**Impact of sampling frequency on cCCM convergence speed: an illustration using sinusoidal waveforms with different frequencies.** (**a**) $f_{0}$ = 75 Hz; (**b**) $f_{0}$ = 100 Hz; (**c**) $f_{0}$ = 200 Hz. Here, $f_{0}$ denotes the frequency of the corresponding sinusoidal waveform. The sampling time sequence was chosen as t=0:0.005:4; that is, sampling frequency $f_{s}=200$ Hz. As can be seen, cCCM works well when the sampling rate is above the Nyquist rate, as shown in (**a**) but may or may not deliver meaningful results when the sampling frequency is below or equal to the Nyquist rate, as shown in (**b**,**c**).

**Figure 7 entropy-26-00539-f007:**
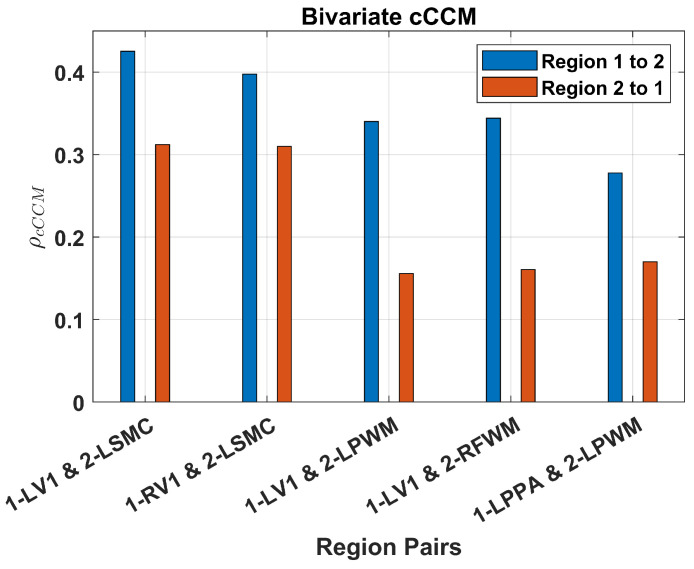
**FMRI-based causality analysis using bivariate cCCM.** Unidirectional causality was detected in the brain network under a visual simulation condition with a scene–object fMRI paradigm.

**Figure 8 entropy-26-00539-f008:**
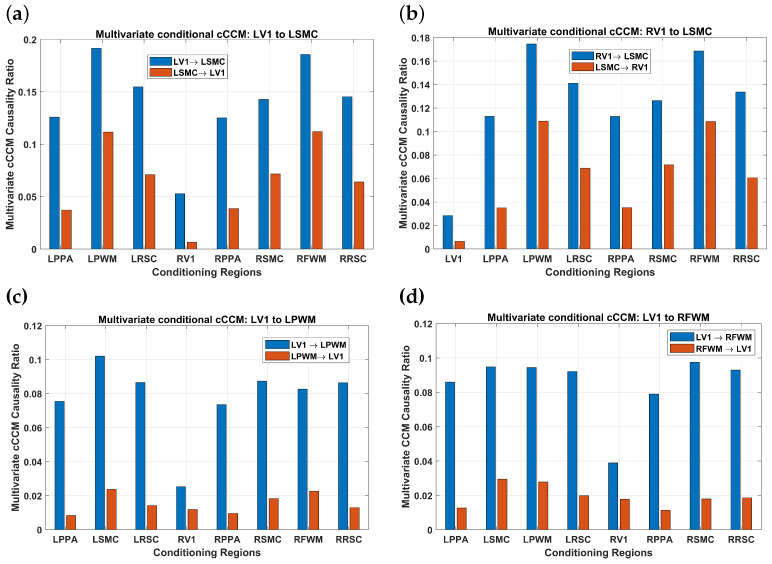
**FMRI-based causality analysis using multivariate conditional cCCM with respect to individual regions.** (**a**) LV1→LSMC, (**b**) RV1→LSMC, (**c**) LV1→LPWM, (**d**) LV1→RFWM. The results indicate that multivariate cCCM (with respect to individual regions) can detect unidirectional causality and also reflect the impact of interdependence between the ROIs on the conditional causality. More specifically, it can be seen that due to the dependence between the brain regions, multivariate conditional CCM values are much smaller than the bivariate cCCM values. In particular, RV1 has the most significant impact on the conditional causality from LV1 → LSMC, LV1→LPWM, and LV1→RFWM. This implies that RV1 has the highest dependence with LV1. For the same reason, LV1 has the the most significant impact on the conditional causality from RV1 → LSMC.

**Figure 9 entropy-26-00539-f009:**
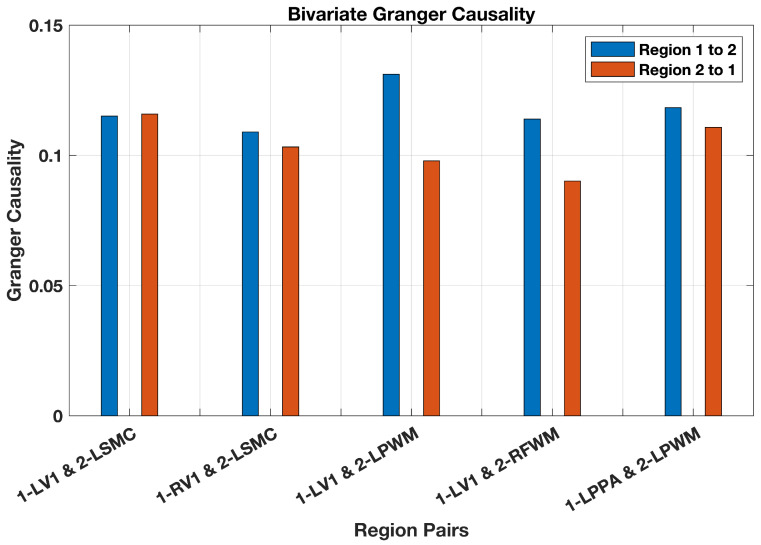
**FMRI-based causality analysis using bivariate GC.** The results of GC are largely consistent with those of bivariate cCCM except from LV1→LSMC. This may be because (i) cCCM takes the instantaneous information exchange between X(t) and Y(t) into consideration, but GC does not; and (ii) cCCM can capture both linear and nonlinear causal causal coupling, and GC may have difficulty in detecting nonlinear causality. That is, when there exists instantaneous information exchange and/or a nonlinear causal relationship between X(t) and Y(t), GC may fail to capture the underlying causal coupling accurately.

**Figure 10 entropy-26-00539-f010:**
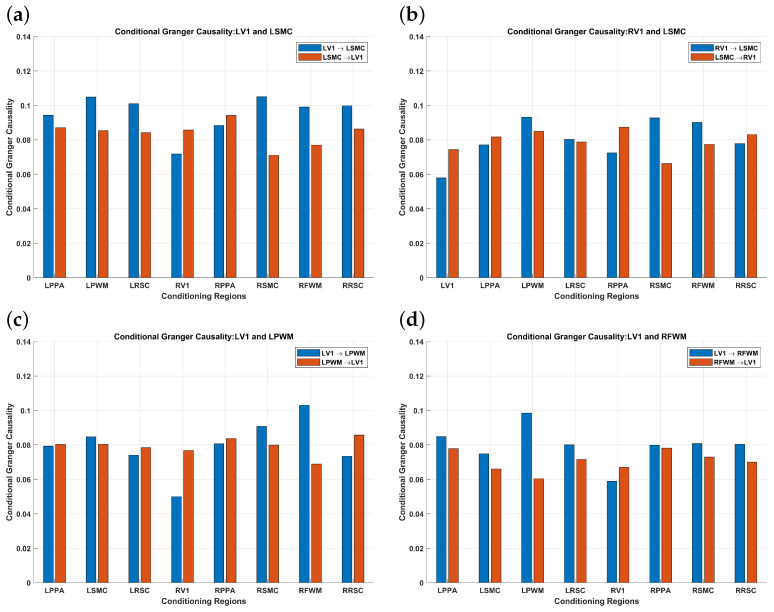
**FMRI-based causality analysis using multivariate conditional GC with respect to individual regions.** (**a**) LV1→LSMC, (**b**) RV1→LSMC, (**c**) LV1→LPWM, (**d**) LV1→RFWM. It can be seen that the results of multivariate cCCM and GC are largely consistent for (**a**,**d**). However, for (**b**,**c**), multivariate cCCM and GC deliver very different results. In particular, for the conditional causality from RV1→LSMC and LV1→LPWM, the results of multivariate cCCM with respect to other individual regions are consistent with their bivariate counterparts and also reflect the impact of inter-region dependence on the conditional causality. However, the corresponding results of multivariate GC with respect to other individual regions vary significantly with the region under consideration, and 50% or more are no longer consistent with those of the bivariate GC.

**Table 1 entropy-26-00539-t001:** **Impact of estimation error** ($n_{1}$ and $n_{2}$ are AWGN noise generated independently of *X* and *Y*, respectively.)

Examples	Direction	$\rho_{\mathrm{cCCM}}$ under Different SNR Values					
		0 dB	5 dB	10 dB	15 dB	20 dB	Noise Free
1. $X_{0}=\mathrm{randn}\left(1000,1\right),$ $X=X_{0}+n_{1}$, $Y=\mathrm{sgn}\left(X_{0}\right)+n_{2}$,	X→Y	0.1896	0.4691	0.6951	0.7819	0.8162	0.8215
	Y→X	0.2598	0.5522	0.6858	0.7133	0.7190	0.7807
	Difference	−0.0701	−0.0830	0.0093	0.0686	0.0972	0.0408
2. $X_{0}=\mathrm{randn}\left(1000,1\right),$ $X=X_{0}+n_{1},$ $Y=X_{0}^{2}+n_{2},$	X→Y	0.1335	0.4332	0.6946	0.8141	0.8460	0.8639
	Y→X	0.0033	0.0197	0.0510	0.1028	0.0473	0.0290
	Difference	0.1302	0.4134	0.6435	0.7113	0.7987	0.8349
3. $X\left(t\right)=\sin\left(t\right)+n_{1},$ $Y\left(t\right)=\cos\left(t\right)+n_{2}.$ t=0:0.01:2π	X→Y	0.1960	0.3234	0.4728	0.6708	0.8917	0.9999
	Y→X	0.3281	0.4981	0.6513	0.7761	0.9080	0.9999
	Difference	−0.1321	−0.1748	−0.1785	−0.1053	−0.0163	0

Here, randn(1000,1) returns a 1000-by-1 matrix of normally distributed random numbers.

**Table 2 entropy-26-00539-t002:** Results for **Example 10**.

Values for *E* and τ	cCCM	CCM
	$\rho_{\mathrm{cCCM}}\left(X\to Y\right):0.0198$	$\rho_{\mathrm{CCM}}\left(X\to Y\right):0.0495$
E=3,τ=1	$\rho_{\mathrm{cCCM}}\left(Y\to X\right):-0.0600$	$\rho_{\mathrm{CCM}}\left(Y\to X\right):0.0087$
	MSE$(X^{n},{\hat{X}}^{n}):1.2416$	MSE$(X^{n},{\hat{X}}^{n}):1.1896$
	MSE$(Y^{n},{\hat{Y}}^{n}):0.9031$	MSE$(Y^{n},{\hat{Y}}^{n}):0.8711$
	$\rho_{\mathrm{cCCM}}\left(X\to Y\right):0.9395$	$\rho_{\mathrm{CCM}}\left(X\to Y\right):0.9535$
E=3,τ=2	$\rho_{\mathrm{cCCM}}\left(Y\to X\right):0.0079$	$\rho_{\mathrm{CCM}}\left(Y\to X\right):0.0630$
	MSE$(X^{n},{\hat{X}}^{n}):1.1876$	MSE$(X^{n},{\hat{X}}^{n}):1.1472$
	MSE$(Y^{n},{\hat{Y}}^{n}):0.0846$	MSE$(Y^{n},{\hat{Y}}^{n}):0.0632$
	$\rho_{\mathrm{cCCM}}\left(X\to Y\right):0.9522$	$\rho_{\mathrm{CCM}}\left(X\to Y\right):0.9776$
E=5,τ=1	$\rho_{\mathrm{cCCM}}\left(Y\to X\right):-0.0210$	$\rho_{\mathrm{CCM}}\left(Y\to X\right):-0.0225$
	MSE$(X^{n},{\hat{X}}^{n}):1.0941$	MSE$(X^{n},{\hat{X}}^{n}):1.0871$
	MSE$(Y^{n},{\hat{Y}}^{n}):0.0868$	MSE$(Y^{n},{\hat{Y}}^{n}):0.0458$

**Table 3 entropy-26-00539-t003:** Results for **Example 11**.

Values for *E* and τ	cCCM	CCM
	$\rho_{\mathrm{cCCM}}\left(X\to Y\right):0.5451$	$\rho_{\mathrm{CCM}}\left(X\to Y\right):0.5833$
E=3,τ=1	$\rho_{\mathrm{cCCM}}\left(Y\to X\right):-0.0521$	$\rho_{\mathrm{CCM}}\left(Y\to X\right):-0.0616$
	MSE$(X^{n},{\hat{X}}^{n}):1.2205$	MSE$(X^{n},{\hat{X}}^{n}):1.2204$
	MSE$(Y^{n},{\hat{Y}}^{n}):0.9445$	MSE$(Y^{n},{\hat{Y}}^{n}):0.8887$
	$\rho_{\mathrm{cCCM}}\left(X\to Y\right):0.5651$	$\rho_{\mathrm{CCM}}\left(X\to Y\right):0.5818$
E=3,τ=2	$\rho_{\mathrm{cCCM}}\left(Y\to X\right):0.0382$	$\rho_{\mathrm{CCM}}\left(Y\to X\right):0.0750$
	MSE$(X^{n},{\hat{X}}^{n}):1.1485$	MSE$(X^{n},{\hat{X}}^{n}):1.1060$
	MSE$(Y^{n},{\hat{Y}}^{n}):0.8996$	MSE$(Y^{n},{\hat{Y}}^{n}):0.8884$
	$\rho_{\mathrm{cCCM}}\left(X\to Y\right):0.9496$	$\rho_{\mathrm{CCM}}\left(X\to Y\right):0.9762$
E=5,τ=1	$\rho_{\mathrm{cCCM}}\left(Y\to X\right):-0.0113$	$\rho_{\mathrm{CCM}}\left(Y\to X\right):0.0123$
	MSE$(X^{n},{\hat{X}}^{n}):1.0797$	MSE$(X^{n},{\hat{X}}^{n}):1.0597$
	MSE$(Y^{n},{\hat{Y}}^{n}):0.1742$	MSE$(Y^{n},{\hat{Y}}^{n}):0.0906$

**Table 4 entropy-26-00539-t004:** The results of **Example 12**.

Values for *E* and τ	cCCM	CCM
	$\rho_{\mathrm{cCCM}}\left(X\to Y\right):0.5373$	$\rho_{\mathrm{CCM}}\left(X\to Y\right):0.5819$
E=3,τ=1	$\rho_{\mathrm{cCCM}}\left(Y\to X\right):-0.0490$	$\rho_{\mathrm{CCM}}\left(Y\to X\right):-0.0151$
	MSE$(X^{n},{\hat{X}}^{n}):1.2765$	MSE$(X^{n},{\hat{X}}^{n}):1.2074$
	MSE$(Y^{n},{\hat{Y}}^{n}):0.9430$	MSE$(Y^{n},{\hat{Y}}^{n}):0.8876$
	$\rho_{\mathrm{cCCM}}\left(X\to Y\right):0.9696$	$\rho_{\mathrm{CCM}}\left(X\to Y\right):0.9910$
E=3,τ=2	$\rho_{\mathrm{cCCM}}\left(Y\to X\right):-0.0184$	$\rho_{\mathrm{CCM}}\left(Y\to X\right):-0.0416$
	MSE$(X^{n},{\hat{X}}^{n}):1.2204$	MSE$(X^{n},{\hat{X}}^{n}):1.2342$
	MSE$(Y^{n},{\hat{Y}}^{n}):0.0854$	MSE$(Y^{n},{\hat{Y}}^{n}):0.0266$
	$\rho_{\mathrm{cCCM}}\left(X\to Y\right):0.9471$	$\rho_{\mathrm{CCM}}\left(X\to Y\right):0.9762$
E=5,τ=1	$\rho_{\mathrm{cCCM}}\left(Y\to X\right):-0.0052$	$\rho_{\mathrm{CCM}}\left(Y\to X\right):0.0088$
	MSE$(X^{n},{\hat{X}}^{n}):1.0647$	MSE$(X^{n},{\hat{X}}^{n}):1.0573$
	MSE$(Y^{n},{\hat{Y}}^{n}):0.1802$	MSE$(Y^{n},{\hat{Y}}^{n}):0.0913$

**Table 5 entropy-26-00539-t005:** The results of **Example 13**.

Values for *E* and τ	cCCM	CCM
	$\rho_{\mathrm{cCCM}}\left(X\to Y\right):0.7900$	$\rho_{\mathrm{CCM}}\left(X\to Y\right):0.8242$
E=5,τ=1	$\rho_{\mathrm{cCCM}}\left(Y\to X\right):-0.0054$	$\rho_{\mathrm{CCM}}\left(Y\to X\right):-0.0016$
	MSE$(X^{n},{\hat{X}}^{n}):1.0799$	MSE$(X^{n},{\hat{X}}^{n}):1.0736$
	MSE$(Y^{n},{\hat{Y}}^{n}):0.5641$	MSE$(Y^{n},{\hat{Y}}^{n}):0.4796$
	$\rho_{\mathrm{cCCM}}\left(X\to Y\right):0.4700$	$\rho_{\mathrm{CCM}}\left(X\to Y\right):0.4972$
E=3,τ=2	$\rho_{\mathrm{cCCM}}\left(Y\to X\right):-0.0197$	$\rho_{\mathrm{CCM}}\left(Y\to X\right):-0.0194$
	MSE$(X^{n},{\hat{X}}^{n}):1.1890$	MSE$(X^{n},{\hat{X}}^{n}):1.1829$
	MSE$(Y^{n},{\hat{Y}}^{n}):1.1990$	MSE$(Y^{n},{\hat{Y}}^{n}):1.1764$
	$\rho_{\mathrm{cCCM}}\left(X\to Y\right):0.9388$	$\rho_{\mathrm{CCM}}\left(X\to Y\right):0.9694$
E=6,τ=1	$\rho_{\mathrm{cCCM}}\left(Y\to X\right):-0.0300$	$\rho_{\mathrm{CCM}}\left(Y\to X\right):-0.0194$
	MSE$(X^{n},{\hat{X}}^{n}):1.0425$	MSE$(X^{n},{\hat{X}}^{n}):1.0419$
	MSE$(Y^{n},{\hat{Y}}^{n}):0.2551$	MSE$(Y^{n},{\hat{Y}}^{n}):0.1462$

**Table 6 entropy-26-00539-t006:** The results of time-delayed causality analysis in **Example 13**.

Values for *E* and τ	cCCM	CCM
	$\rho_{\mathrm{cCCM}}\left(X\to Y_{1}\right):0.9490$	$\rho_{\mathrm{CCM}}\left(X\to Y_{1}\right):0.9753$
E=5,τ=1	$\rho_{\mathrm{cCCM}}\left(Y_{1}\to X\right):0.5947$	$\rho_{\mathrm{CCM}}\left(Y_{1}\to X\right):0.6380$
	MSE$(X^{n},{\hat{X}}^{n}):0.5841$	MSE$(X^{n},{\hat{X}}^{n}):0.5362$
	MSE$(Y_{1}^{n},{\hat{Y}}_{1}^{n}):0.2032$	MSE$(Y_{1}^{n},{\hat{Y}}_{1}^{n}):0.1061$
	$\rho_{\mathrm{cCCM}}\left(X\to Y_{1}\right):0.6621$	$\rho_{\mathrm{CCM}}\left(X\to Y_{1}\right):0.6854$
E=3,τ=2	$\rho_{\mathrm{cCCM}}\left(Y_{1}\to X\right):0.5309$	$\rho_{\mathrm{CCM}}\left(Y_{1}\to X\right):0.5683$
	MSE$(X^{n},{\hat{X}}^{n}):0.6814$	MSE$(X^{n},{\hat{X}}^{n}):0.6404$
	MSE$(Y_{1}^{n},{\hat{Y}}_{1}^{n}):0.8317$	MSE$(Y_{1}^{n},{\hat{Y}}_{1}^{n}):0.7992$
	$\rho_{\mathrm{cCCM}}\left(X\to Y_{1}\right):0.9421$	$\rho_{\mathrm{CCM}}\left(X\to Y_{1}\right):0.9705$
E=6,τ=1	$\rho_{\mathrm{cCCM}}\left(Y_{1}\to X\right):0.5861$	$\rho_{\mathrm{CCM}}\left(Y_{1}\to X\right):0.6405$
	MSE$(X^{n},{\hat{X}}^{n}):0.5874$	MSE$(X^{n},{\hat{X}}^{n}):0.5332$
	MSE$(Y_{1}^{n},{\hat{Y}}_{1}^{n}):0.2555$	MSE$(Y_{1}^{n},{\hat{Y}}_{1}^{n}):0.1450$

**Table 7 entropy-26-00539-t007:** The results of **Example 14**.

Values for *E* and τ	cCCM	CCM
	$\rho_{\mathrm{cCCM}}\left(X\to Y\right):0.6947$	$\rho_{\mathrm{CCM}}\left(X\to Y\right):0.7286$
E=5,τ=1	$\rho_{\mathrm{cCCM}}\left(Y\to X\right):-0.0250$	$\rho_{\mathrm{CCM}}\left(Y\to X\right):-0.0024$
	MSE$(X^{n},{\hat{X}}^{n}):1.0910$	MSE$(X^{n},{\hat{X}}^{n}):1.0849$
	MSE$(Y^{n},{\hat{Y}}^{n}):0.4670$	MSE$(Y^{n},{\hat{Y}}^{n}):0.4287$
	$\rho_{\mathrm{cCCM}}\left(X\to Y\right):0.7019$	$\rho_{\mathrm{CCM}}\left(X\to Y\right):0.7309$
E=3,τ=2	$\rho_{\mathrm{cCCM}}\left(Y\to X\right):-0.0327$	$\rho_{\mathrm{CCM}}\left(Y\to X\right):0.0119$
	MSE$(X^{n},{\hat{X}}^{n}):1.1890$	MSE$(X^{n},{\hat{X}}^{n}):1.1829$
	MSE$(Y^{n},{\hat{Y}}^{n}):1.2170$	MSE$(Y^{n},{\hat{Y}}^{n}):1.1760$
	$\rho_{\mathrm{cCCM}}\left(X\to Y\right):0.9479$	$\rho_{\mathrm{CCM}}\left(X\to Y\right):0.9718$
E=6,τ=1	$\rho_{\mathrm{cCCM}}\left(Y\to X\right):-0.0300$	$\rho_{\mathrm{CCM}}\left(Y\to X\right):-0.0194$
	MSE$(X^{n},{\hat{X}}^{n}):1.0425$	MSE$(X^{n},{\hat{X}}^{n}):1.0419$
	MSE$(Y^{n},{\hat{Y}}^{n}):0.2551$	MSE$(Y^{n},{\hat{Y}}^{n}):0.1462$

**Table 8 entropy-26-00539-t008:** The results of time-delayed causality analysis in **Example 14**.

Values for *E* and τ	cCCM	CCM
	$\rho_{\mathrm{cCCM}}\left(X\to Y_{3}\right):0.9543$	$\rho_{\mathrm{CCM}}\left(X\to Y_{3}\right):0.9782$
E=5,τ=1	$\rho_{\mathrm{cCCM}}\left(Y_{3}\to X\right):-0.0378$	$\rho_{\mathrm{CCM}}\left(Y_{3}\to X\right):-0.0095$
	MSE$(X^{n},{\hat{X}}^{n}):1.0968$	MSE$(X^{n},{\hat{X}}^{n}):1.0736$
	MSE$(Y_{3}^{n},{\hat{Y}}_{3}^{n}):0.1210$	MSE$(Y_{3}^{n},{\hat{Y}}^{n}):0.0636$
	$\rho_{\mathrm{cCCM}}\left(X\to Y_{3}\right):0.4445$	$\rho_{\mathrm{CCM}}\left(X\to Y_{3}\right):0.4568$
E=3,τ=2	$\rho_{\mathrm{cCCM}}\left(Y\to X\right):-0.0606$	$\rho_{\mathrm{CCM}}\left(Y_{3}\to X\right):-0.0611$
	MSE$(X^{n},{\hat{X}}^{n}):1.2451$	MSE$(X^{n},{\hat{X}}^{n}):1.2306$
	MSE$(Y^{n},{\hat{Y}}_{3}^{n}):0.7748$	MSE$(Y_{3}^{n},{\hat{Y}}_{3}^{n}):0.7743$
	$\rho_{\mathrm{cCCM}}\left(X\to Y_{3}\right):0.9424$	$\rho_{\mathrm{CCM}}\left(X\to Y_{3}\right):0.9703$
E=6,τ=1	$\rho_{\mathrm{cCCM}}\left(Y_{3}\to X\right):-0.0293$	$\rho_{\mathrm{CCM}}\left(Y_{3}\to X\right):-0.0173$
	MSE$(X^{n},{\hat{X}}^{n}):1.0545$	MSE$(X^{n},{\hat{X}}^{n}):1.0432$
	MSE$(Y_{3}^{n},{\hat{Y}}_{3}^{n}):0.1590$	MSE$(Y_{3}^{n},{\hat{Y}}_{3}^{n}):0.0927$

**Table 9 entropy-26-00539-t009:** Results for **Example 15**.

Noise	$n\left(t\right)∼N\left(0,\sigma^{2}\right)$				
	$\sigma^{2}=0$	$\sigma^{2}=10^{-6}$	$\sigma^{2}=10^{-2}$	$\sigma^{2}=10^{-1}$	$\sigma^{2}=4$
SNR(dB)	∞ (noise free)	52.26 dB	12.26 dB	2.26 dB	−7.74 dB
GC(X→Y)	$6.078\times 10^{-5}$	$6.383\times 10^{-4}$	0.0065	0.0345	0.0277
GC(Y→X)	$6.862\times 10^{-5}$	$6.589\times 10^{-4}$	0.0019	0.0040	0.0035
$\rho_{\mathrm{cCCM}}\left(X\to Y\right)$	0.9169	0.9168	0.9070	0.8314	0.1897
$\rho_{\mathrm{cCCM}}\left(Y\to X\right)$	0.9024	0.9023	0.8873	0.8043	0.1413

**Table 10 entropy-26-00539-t010:** Performance of cCCM under additive white Guassian noise with different *E* and τ values.

Values for *E* and τ	Direction	0 dB	5 dB	10 dB	15 dB	20 dB	Noise Free
E=5,τ=1	X→Y	0.2378	0.2410	0.3497	0.5156	0.6273	0.9566
	Y→X	0.2476	0.4446	0.6194	0.7267	0.8447	0.9945
E=5,τ=5	X→Y	0.3497	0.6116	0.8278	0.9432	0.9799	0.9985
	Y→X	0.5570	0.7566	0.8893	0.9693	0.9877	0.9991

## Data Availability

The fMRI datasets presented in this study are available to qualified investigators according to the NIH data sharing policy upon reasonable request. All the other data supporting the findings of this study are available within the article. The relevant MATLAB code can be found at https://github.com/BAWC-Evan-Sun/CCM-Implement.git (accessed on 20 June 2024).

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
