# Peer review of "Causalized Convergent Cross Mapping and Its Implementation in Causality Analysis"

_entropy, 2024, doi:10.3390/e26070539_

Round 1
Reviewer 1 Report
Comments and Suggestions for Authors
This paper introduced the casualized convergent cross mapping (CCM) method for causal inference between time series. The original method does not exclude future signals in the analysis of dependence between time series. This paper modified the method by excluding future signals and examining the importance of the past signals of one-time series in the prediction or another one. In the previous work [9], the team has shown the equivalence between cCCM and the Directed Information (DI) method. Overall the method is interesting to this field. Below are several comments to further improve the quality of this paper.
1) The main contribution of this paper is not clear. Abstract and Introduction claimed that the concept of cCCM is introduced in this paper. But it seems that it was already introduced in [9]. It would be helpful to clarify the contributions.
2) It would be helpful to provide insights about the method in Eq. (3). Why are the coefficients fixed based on the Euclidian distance? The standard GC method learns the joint model of X and Y using an autoregressive model or a more general state-space representation method.
3) The cCCM method is related to GC via the DI method. Is the multivariate cCCM related to the conditional Granger causality measure [Geweke J. Measures of conditional linear dependence and feedback between time series. J Am Stat Assoc. 1984;79:907–915.] There is a paper on an information-theoretic method for multivariate causality analysis: https://doi.org/10.1162/netn_a_00386. It would be helpful to discuss the relations between these methods and multivariate cCCM.
4) All the simulation examples are based on two-variable time series. It would be helpful to demonstrate the multivariate cCCM method. However, it is not mandatory since the real-data example showed the multivariate cCCM method.
Author Response
Dear Reviewer,
Thank you for taking the time to review our manuscript and provide insightful comments. The paper has been revised accordingly. Attached please find our point-to-point response to your comments.
Sincerely,
Boxin Sun

Reviewer 2 Report
Comments and Suggestions for Authors
Thank you for the invitation to review the manuscript "Causalized Convergent Cross Mapping and Its Implementation in Causality Analysis."
I find the work quite interesting and a good fit for the Special Issue. The authors focus on revisiting the original definition of convergent cross mapping (CCM) and its gap with the traditional view of causality, established by Granger's work and other theoretical information metrics such as directed information. They demonstrate how this gap can be resolved by their causalized CCM (cCCM). The paper builds on prior work where these definitions were discussed at length (ref. [9]). After introducing the definitions and extensions, the authors evaluate the effectiveness of cCCM through several examples: Gaussian random variables with additive noise, sinusoidal waveforms, autoregressive models, stochastic processes with a dominant spectral component embedded in noise, deterministic chaotic maps, systems with memory, and fMRI data. Clearly, the authors tested cCCM exhaustively, demonstrating its effectiveness in linear and nonlinear causal coupling.
My major comment is that the authors should clarify what results are specific to this paper and what methods are derived from their previous publications. Many parts of sections 2.1, 2.2, and even 2.3 seem to be a review of the model, and should not be included in the results section. Alternatively, the authors should clearly indicate the extensions that are unique to this paper.
My minor comment, more of a suggestion, is that the authors should provide better documentation of the code and examples. Currently, it is difficult to understand and use cCCM due to the nearly absent documentation and limited scripts provided. An easy-to-follow example with a well-documented .md file would be very helpful for others to use the same algorithms.
Additionally, I wonder if this approach could be applied to point processes, such as spikes (action potentials) and the identification of causal relationships (connections between neurons).
Author Response

(The authors gave the same response as above.)

Round 2
Reviewer 1 Report
Comments and Suggestions for Authors
The revision has addressed most of my previous comments and concerns. However, I do have some suggestions for the new Example 15. It was claimed in this example that the GC method is not sensitive to the instantaneous dependence of the variables. However, the results in Example 15 are also related to the fact that the input noise n(t) to y(t) is almost singular. My numerical experiment has shown that by increasing the variance of n(t), the GC(x to y) will be increased. Thus, I think the singularity of the input covariance matrix may be another factor that the GC measure did not provide correct results.
Author Response
Dear Reviewer,
Thank you for reviewing the revised manuscript and providing the new comments. Attached please find our point-to-point response.
Sincerely,
Boxin Sun

Reviewer 2 Report
Comments and Suggestions for Authors
I appreciate the efforts made by the authors to improve the manuscript. The manuscript has been improved considerably, and the addition of more examples is appreciated. I believe it can be accepted.
Author Response
Dear Reviewer,
Thank you for reviewing the revised manuscript and for your support of our research! Attached please find our point-to-point response to the comments from the reviewers.
Sincerely,
Boxin Sun
